# Refinement of pore size at sub-angstrom precision in robust metal–organic frameworks for separation of xylenes

Xiaolin Li[1,2], Juehua Wang[1], Nannan Bai[1], Xinran Zhang [1], Xue Han[1], Ivan da Silva [3], Christopher G. Morris[1], Shaojun Xu [1], Damian M. Wilary[1], Yinyong Sun [2], Yongqiang Cheng[4], Claire A. Murray[5], Chiu C. Tang[5], Mark D. Frogley [5], Gianfelice Cinque[5], Tristan Lowe[6], Haifei Zhang[7], Anibal J. Ramirez-Cuesta [4], K. Mark Thomas [8], Leslie W. Bolton[9], Sihai Yang [1✉] & Martin Schröder [1✉]

The demand for xylenes is projected to increase over the coming decades. The separation of xylene isomers, particularly *p*- and *m*-xylenes, is vital for the production of numerous polymers and materials. However, current state-of-the-art separation is based upon fractional crystallisation at 220 K which is highly energy intensive. Here, we report the discrimination of xylene isomers via refinement of the pore size in a series of porous metal–organic frameworks, MFM-300, at sub-angstrom precision leading to the optimal kinetic separation of all three xylene isomers at room temperature. The exceptional performance of MFM-300 for xylene separation is confirmed by dynamic ternary breakthrough experiments. In-depth structural and vibrational investigations using synchrotron X-ray diffraction and terahertz spectroscopy define the underlying host–guest interactions that give rise to the observed selectivity (*p*-xylene < *o*-xylene < *m*-xylene) and separation factors of 4.6–18 for *p*- and *m*-xylenes.

[1] Department of Chemistry, University of Manchester, Manchester M13 9PL, UK. [2] School of Chemistry and Chemical Engineering, Harbin Institute of Technology, Harbin 150001, China. [3] ISIS Facility, STFC Rutherford Appleton Laboratory, Oxfordshire OX11 0QX, UK. [4] Neutron Scattering Division, Neutron Sciences Directorate, Oak Ridge National Laboratory, Oak Ridge, TN 37831, USA. [5] Diamond Light Source, Harwell Science Campus, Oxfordshire OX11 0DE, UK. [6] Henry Moseley X-ray Imaging Facility, Photon Science Institute, University of Manchester, Manchester M13 9PL, UK. [7] Department of Chemistry, University of Liverpool, Liverpool L69 7ZD, UK. [8] School of Chemical Engineering and Advanced Materials, Newcastle University, Newcastle upon Tyne NE1 7RU, UK. [9] BP Group Research, Sunbury-upon-Thames TW16 7BP, UK. ✉email: sihai.yang@manchester.ac.uk; m.schroder@manchester.ac.uk

The chemical separation of valuable feedstocks based upon fractional crystallisation or distillation is being carried out on a vast scale worldwide[1]. These processes account for 10–15% of the world's energy consumption and up to 70% of the running-costs of chemical plants[1,2]. There is a continual search for alternative separation technologies operating under ambient conditions that can, over time, replace the current processes to reduce energy consumption[3]. This requires the design of smart functional materials that can discriminate small molecules based upon slight differences in molecular structures and/or physical properties. The separation of xylene isomers is regarded as one of the seven world-challenging separations[1].

The supply chains for many polymers, plastics, fibres, textiles, solvents and fuel additives rely heavily upon delivery of pure xylene isomers[4,5]. For example, p-xylene is used in the manufacture of poly(ethylene terephthalate) (PET), whereas o-xylene is used to produce phthalic anhydride[6,7]. m-Xylene can be oxidised to isophthalic acid as PET resin blends, and, more importantly, isomerised into value-added p-xylene[8]. Commercial xylene is composed of m-xylene (40–65%), p-xylene (~20%) and o-xylene (~20%)[9]. Owing to the similarity of the boiling points of p-xylene (138.4 ℃), m-xylene (139 ℃) and o-xylene (144 ℃), their separation in industry is currently performed via either fractional crystallisation operating at ~220 K[3] or adsorption in porous adsorbents, such as cation-exchanged FAU-type zeolites X and Y[6,10]. In the former process, the recovery of p-xylene is limited to 60–70% due to the eutectic point accompanied with the high energy consumption utilised for cooling. In adsorption-based processes, such as the commercial Parex process[11], high temperature (~180 °C) and pressure (~9 bar) are required to promote the separation[12]. Improving the separation efficiency between xylene isomers in zeolites is not a trivial task, not least because of their very rigid pore structure and hence the lack of design flexibility[13]. Therefore, there is a significant incentive to develop new functional sorbents that can better discriminate between xylene isomers, particularly p- and m-isomers; m-xylene accounts for over half of commercial xylenes and has a very similar boiling point to the high-value p-xylene.

Porous metal–organic frameworks (MOFs) have been widely studied over the past two decades owing to their high internal surface area[14,15] and, more importantly, tailored-to-application structural tuneability[16–20]. In exceptional cases, MOFs can show emerging potential for xylene separations, particularly using flexible MOFs[21–28]. For example, MIL-53(Al) is selective for o-xylene with a selectivity (S) of 2.2 at 298 K, but is unable to discriminate between p- and m-xylene ($S_{p/m} = 1.0$)[21]. Likewise, MIL-47(V) shows a high $S_{p/m}$ of 2.0 at 298 K but again shows low selectivity for p-xylene over o-xylene ($S_{p/o} = 1.0$)[27]. Using the analogous MIL-53(Fe), the full separation of xylenes has been achieved with an elution order of p-, m- and then o-xylene albeit with moderate separation factors (α) at 323 K ($\alpha_{m/p} \sim 1.5$)[22]. In comparison, a rigid MOF, MIL-125(Ti)-NH$_2$, exhibits a high selectivity for p-xylene with a $S_{p/m}$ of 3.0, but poor separation between o-xylene and m-xylene ($S_{o/m} = 1.0$)[28], as observed also for pillar(n)arene materials[29]. Here, we demonstrate a molecular discrimination mechanism for xylene isomers via refining the pore size, at a sub-angstrom precision, in a series of rigid MOFs, MFM-300(M) (M = Al, Fe, V, In). The full separation of xylenes into three pure isomers is achieved in MFM-300 through both chromatographic and breakthrough experiments at room temperature. Importantly, MFM-300 materials show a unique elution sequence for xylene isomers (p-, o- and then m-xylene) that is distinct to previously reported MOFs. Indeed, MFM-300 exhibits exceptional selectivities towards m-xylene ($S_{m/p} = 3.8$ and 3.9 in MFM-300(In) and MFM-300(Fe), respectively). This is highly desirable for practical processes as it enables (i) rapid elution of the most valuable p-isomer; (ii) selective retention of m-xylene to feed into subsequent isomerisation processes; (iii) the optimised separation of p-xylene and m-xylene. We also report the direct observation of binding domains and host–guest interactions for adsorbed xylene molecules in the pores of MFM-300 using a combination of high-resolution synchrotron X-ray powder diffraction, terahertz spectroscopy and molecular dynamic (MD) modelling. These complementary techniques afford direct evidence for the formation of distinct host–guest interactions as a function of pore size that underpin the unique selectivity observed in MFM-300 materials.

## Results

**Synthesis and characterisation of MFM-300.** The four iso-structural MOFs, MFM-300(M) (M = In, V, Fe, Al), were selected for this study owing to their robust open structures comprising one-dimensional square-shaped channels with periodic arrangements of bridging hydroxyls and phenol rings, which can interact with guest molecules in a selective manner[30,31]. These MOFs exhibit well-defined pore sizes, i.e., 7.4, 7.0, 6.8, and 6.5 Å for MFM-300(M) (M = In, V, Fe, Al), respectively, owing to the differences in the radii of the M(III) centres and M-ligand distances. Throughout this report, all discussion of pore size takes into account the van der Waals radii of surface atoms. Thus, MFM-300(M) (M = Al, Fe, V, In) provide an excellent opportunity to probe a narrow pore size region between 6.5 and 7.4 Å relevant to molecular cross-section sizes of p-, o-, and m-xylene with the potential to identify the "sweet spot" for their separation (Fig. 1). Specifically, MFM-300(M) (M = In, V, Fe) have channel sizes comparable to the diameter of xylene isomers, whereas the pore size for MFM-300(Al) is narrower than m- and o-xylene. A detailed structural analysis revealed that in each MFM-300 material the pore dimension decreases periodically by 0.2 Å at the hydroxyl bridges, affording a helical zigzag 1D channel (Fig. 1). For example, the pore dimension varies between 7.4 and 7.2 Å periodically along the c axis in MFM-300(In). This minor variation in pore size can potentially promote separations via a kinetic molecular sieving mechanism, particularly for xylene isomers with similar molecular sizes (Table 1). The geometry of the different pores in these materials is shown in Supplementary Fig. 1, with the overall connectivity affording a porous 3D framework structure with square-shaped channels. The phase purity of all MOFs and their stability towards xylenes have been confirmed by PXRD data (Supplementary Fig. 2). There is no detectable change in either crystallinity or porosity of the four samples after exposure to and separation of xylenes (Supplementary Figs. 2 and 3). The BET surface areas of all used samples were determined to be ~1000 m$^2$ g$^{-1}$ after xylene separation (Supplementary Table 1). The pore size distributions of MFM-300 samples determined from isotherm data (Supplementary Fig. 4) are consistent with the calculated values (Fig. 1), whereas TGA measurements confirm thermal decomposition of these MOF materials at around 350 °C (Supplementary Fig. 5).

**Assembly and imaging studies of high performance liquid chromatography (HPLC) columns packed with MFM-300.** Scanning electron microscopy (SEM) confirms that all MFM-300 samples adopt a prismatic rod shape (Supplementary Fig. 6). The morphology and uniformity of particles are retained upon grinding in a mortar (Supplementary Fig. 7). Powder samples of MFM-300(M) (M = In, V, Fe, Al) were packed into stainless steel columns (15 cm length; 4.6 mm i.d.) under high pressure to minimise the presence of voids, which are detrimental to the column efficiency. The column packed with MFM-300(In) has been further studied by 3D X-ray imaging (Supplementary Fig. 8).

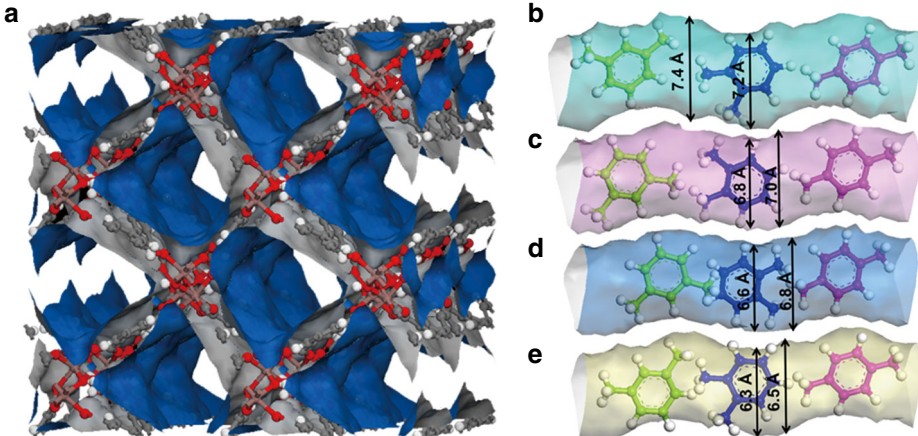

**Fig. 1 Views of crystal structures of MFM-300. a** Projected view along the *c* axis and views of the one-dimensional channels in MFM-300, (metal, purple; carbon, grey; oxygen, red; hydrogen, white), **b** views of MFM-300(In), **c** MFM-300(V), **d** MFM-300(Fe), **e** MFM-300(Al). The pore dimensions were obtained using Material Studio software based on the CIFs of MFM-300 determined from X-ray diffraction experiments. The molecular structures for three xylene isomers are also included for comparison with the pore dimensions. The pore sizes are: MFM-300(Al) (6.5–6.3 Å), MFM-300(Fe) (6.8–6.6 Å), MFM-300(V) (7.0–6.8 Å), MFM-300(In) (7.4–7.2 Å). MFM-300(Al) has a slightly narrower pore than the dimensions of *o*- and *m*-xylene isomers, as illustrated.

**Table 1 Comparison of the chromatographic separation factors (α) for xylene isomers in different MOFs.**

| MOFs | Pore dimensions (Å) | Temperature (K) | $\alpha_{o/p}$ | $\alpha_{m/o}$ | $\alpha_{m/p}$ |
|---|---|---|---|---|---|
| MFM-300(In) | 7.4 | 293 | 1.6 | 2.9 | 4.6 |
| MFM-300(V) | 7.0 | 293 | 5 | 3 | 15 |
| MFM-300(Fe) | 6.8 | 293 | 3 | 6 | 18 |
| MFM-300(Al) | 6.5 | 293 | – | 1.0 | – |
| MFM-300(In)(V) | – | 293 | 2.4 | 2.8 | 6.7 |
| MIL-125(Ti)-NH$_2$[28] | 10.7 | 298 | 0.45[a] | 1.0[a] | 0.33[a] |
| UiO-66(Zr)[32] | 11 | 313 | 1.4 | 0.8 | 1.1 |
| MIL-53(Fe)[22] | 8.4 | 293 | 3.5 | 0.8 | 2.6 |
| MIL-47(V)[34] | 7.9 | 298 | 1.4[b] | 0.5[b] | 0.4[b] |
| MIL-53(Al)ht[34] | 8.5 | 298 | 3.5[b] | 0.37[b] | 1.2[b] |
| MIL-101(Cr)[35] | 22 | 293 | 2.7 | 0.54 | 1.5 |

[a]Calculated from binary breakthrough curves.
[b]Calculated from binary batch uptake experiments (0.028 M).

This confirms the absence of macro-voids at the sub-micron level and thus very compact packing of the MOF sample can be achieved within the entire column, ideal for applications in dynamic separations. It is worth noting that the edges of MOF particles are not visible in the X-ray images owing to the dense packing of the powder.

**HPLC separation of equimolar *o*, *m* and *p*-xylene isomers.** The retention of xylene isomers in MFM-300 was determined by HPLC experiments at room temperature, where a fixed amount of equimolar mixture (1:1:1) of *o*-, *m*- and *p*-xylenes was injected into the column and the elution monitored by GC continuously until all injected xylenes were recovered from the exhaust (Fig. 2). The effect of flow rate and eluent on xylene separation have been systematically tested (Supplementary Figs. 9 and 10). *n*-Pentane was selected as the eluent owing to its lower boiling point (36.1 °C) than xylenes and thus it can be readily recycled. MFM-300(In) with the largest pore size (7.4–7.2 Å) in this series was tested first. The order of elution is *p*-, *o*- and then *m*-xylene and, importantly, *m*-xylene was preferentially retained over the other two isomers and can be completely isolated from MFM-300(In). With slightly reduced pores (7.0–6.8 Å), MFM-300(V) and MFM-300(Fe) both show selectivity for *m*-xylene and exhibit the same

order of elution (i.e., *p*-, *o*- and then *m*-xylene). However, when the pore is further reduced to 6.5–6.3 Å, MFM-300(Al) displays poor activity for the separation of xylenes under the same conditions. The bed void and retention times in different columns are shown in Supplementary Fig. 11 and Supplementary Table 2, respectively. The separation factors ($\alpha_{i,j}$; where *i* and *j* stand for two components in the mixture) have been calculated from the chromatographic data and compared with the best-behaving MOFs reported to date (Table 1). Significantly, exceptional separation factors of $\alpha_{m,p}$ have been obtained for MFM-300(V) and MFM-300(Fe) (15 and 18, respectively), significantly exceeding those reported for the state-of-the-art MOFs, such as 2.6, 1.1 and 2.7 for MIL-53(Fe)[22], UiO-66(Zr)[32] and CD-MOF[33], respectively. Interestingly, with the best peak resolution in the chromatogram, MFM-300(In) and MFM-300(V) show complementary selectivities: MFM-300(V) is more efficient for the *o*- and *p*-xylene separation, whereas MFM-300(In) can completely differentiate between *m*-xylene and *o*-xylene (resolution factor $R_{m/o} = 1.7$). Thus, theoretically, a baseline separation of all three xylene isomers could be achieved by a tandem connection of these two columns. Importantly and significantly, the near complete separation of equimolar mixture of *o*-, *m*- and *p*-xylenes with combined optimal separation factors and resolution ($R_{m/o} = 1.5$ and $R_{o/p} = 1.1$) has been observed from an integrated column

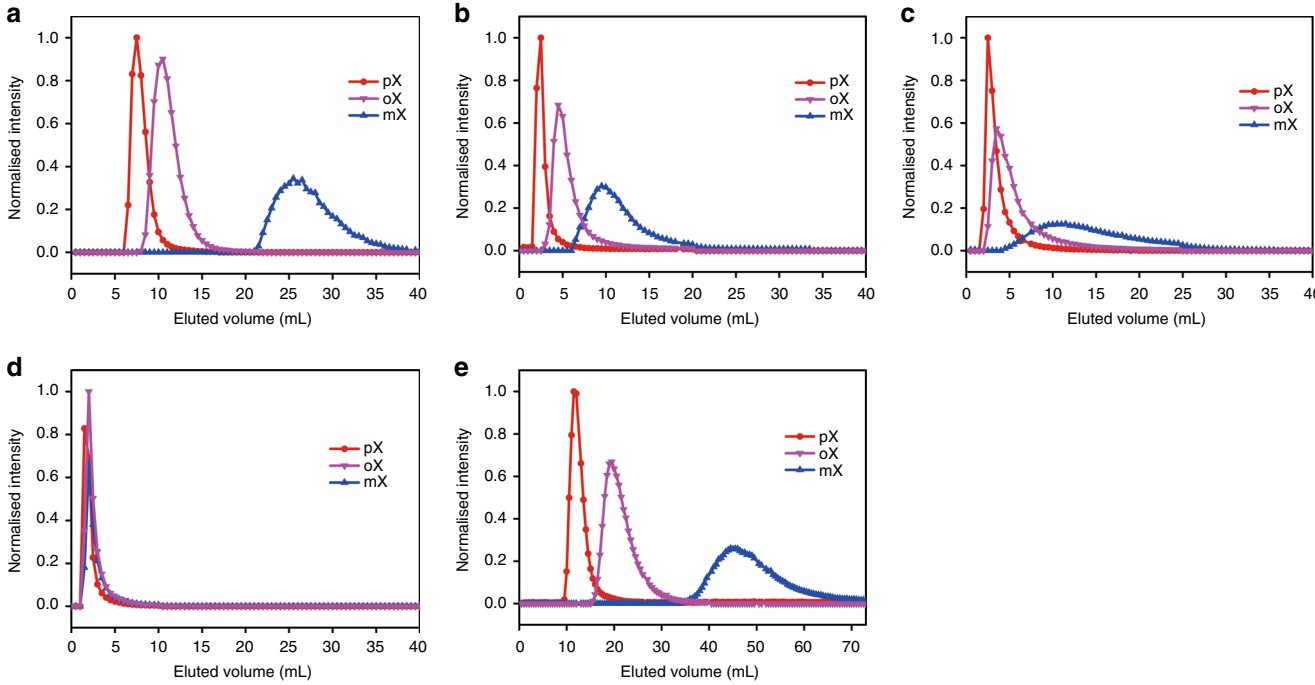

**Fig. 2 Chromatographic separation of equimolar mixture of *o*-, *m*- and *p*-xylene in different columns. a** MFM-300(In), **b** MFM-300(V), **c** MFM-300 (Fe), **d** MFM-300 (Al) and **e** two columns in series comprising MFM-300(In) and MFM-300(V). Measurements were undertaken at 293 K.

**Table 2 Summary of physical properties of three xylene isomers[4,9].**

| Xylene isomer | *o*-xylene | *m*-xylene | *p*-xylene |
|---|---|---|---|
| Molecular structure | | | |
| Boiling point (°C) | 144.4 | 139.1 | 138.4 |
| Melting point (°C) | −25 | −48 | 13 |
| Molecular dimension (Å)[a] | 7.269 × 3.834 × 7.826 | 8.994 × 3.949 × 7.315 | 6.618 × 3.810 × 9.146 |
| MIN-1(Å) | 3.834 | 3.949 | 3.810 |
| MIN-2(Å) | 7.269 | 7.258 | 6.618 |
| MIN-1×MIN-2(Å$^2$) | 27.869 | 28.662 | 25.215 |
| Polarizability (×10$^{−25}$ cm$^3$) | 141–149 | 142 | 137–149 |
| Composition in commercial xylenes (%) | ~20 | 40–65 | ~20 |

[a]Taken from Webster, C.E.; Drago, R.S.; Zerner, M.C. Molecular dimensions for adsorptives, *J. Am. Chem. Soc.* **120**, 5509–5516 (1998).

of MFM-300(In) and MFM-300(V) (Fig. 2e). The sequence of the tandem coupling was found to have no detectable effect on the separation.

The elution sequence of three xylene isomers from MFM-300 (M) (M = In, V, Fe) followed the same order, and this is distinct to all previously reported MOFs[21–23,27,33–37] and COFs[38], where varying elution orders were observed in different systems, even for the iso-structural MIL-53/MIL-47 systems[21,34]. When considering adsorption characteristics, in the case of an adsorbent with slit-shaped pores the minimum molecular cross-section size of the molecule in one dimension is critical. In the case of the adsorbent with cylindrical pores, the minimum circular cross-section of the molecule in two dimensions needs to be considered[39]. The dimensions of xylene isomers are compared in Table 2. MFM-300 exhibits square/circular-shaped pores, and thus two minimum molecular dimensions must be considered for diffusion of xylene molecules through the porous structure. The

molecular dimensions MIN-1 and MIN-2 of xylene molecules follow an increasing order of *p*-xylene = 3.810 × 6.618 Å < *o*-xylene = 3.834 × 7.269 Å < *m*-xylene = 3.949 × 7.258 Å. In the case of kinetic molecular sieving, a decrease in diffusion rate would be expected with increasing size of the substrate. This is in excellent agreement with the elution order observed in MFM-300 (M) (M = In, V, Fe), and thus it is evident that the diffusion of xylene isomers along the channels of MFM-300 is vital to the observed separation process.

**Breakthrough separations of equimolar *o*-, *m*- and *p*-xylenes.** Separation processes in industry are typically operated under flow conditions. We sought to test the dynamic separation of xylenes in MFM-300(M) (M = In, V, Fe) via binary breakthrough experiments (Supplementary Figs. 12 and 13), which enabled the determination of sorption selectivities in liquid phase

(Supplementary Table 3). The materials MFM-300(M) (M = In, V, Fe) showed the same order of breakthrough, i.e., *p*-, *o*- and then *m*-xylene, consistent with the chromatographic experiments. MFM-300(V) exhibited a high selectivity of 3.0 for *o*- vs *p*-xylene, and a poor selectivity of 1.1 for *m*- vs *o*-xylene. In contrast, MFM-300(In) shows a poor separation of *o*- vs *p*-xylene (S$_{o/p}$ = 1.3), but a high selectivity of 3.5 for *m*- vs *p*-xylene. This result is in excellent agreement with the chromatographic results, confirming their complementary capability for dynamic xylene separation. A "roll-up" effect[40,41] has been observed in all the three adsorbents studied, and is synchronous with the breakthrough of *m*-xylene, indicating that the most strongly adsorbed *m*-xylene can displace the initially adsorbed *p*-xylene and *o*-xylene from the pores, indicating the presence of strong binding of *m*-xylene in MFM-300. As ethylbenzene is also a bi-product of catalytic reforming of crude oil, binary breakthrough separations of ethylbenzene and xylenes have also been tested for MFM-300(M) (M = In, V, Fe) (Supplementary Fig. 13). All three MOFs can distinguish ethylbenzene from the three xylene isomers. MFM-300(M) (M = In, Fe) showed the same order of breakthrough, i.e., *p*-xylene, ethylbenzene, *o*- and then *m*-xylene. In contrast, MFM-300(V) showed a slightly different elution sequence of *p*-, *o*-, ethylbenzene and *m*-xylene with poor discrimination between *o*-xylene and ethylbenzene.

To further probe the potential of MFM-300 for practical xylene separation, highly challenging ternary breakthrough experiments were conducted using an equimolar mixture of three xylene isomers (Fig. 3 and Supplementary Fig. 14). By pumping a feed containing an equimolar mixture of 15 mM and 50 mM over the column, respectively, all materials show clear separations and the same breakthrough order (i.e., *p*- < *o*- < *m*-xylene) as above. Similar "roll-up" effect was also observed, indicating the three xylene isomer molecules are competitively adsorbed in the pores and also, there is competition between the xylenes and *n*-pentane. The breakthrough times maintained when using single-component solutions. However,

the "roll-up" effect disappeared, further confirming that the xylene molecules in mixtures compete for the same space within the pores. Selectivity values have been calculated from these ternary breakthrough curves and compared with the leading MOFs to date (Fig. 4). Remarkably, MFM-300(In), MFM-300(Fe) and the tandem column MFM-300(M) (M = In, V) show excellent selectivities with a record-high *m*- vs *p*-xylene selectivity (S$_{m/p}$ = 3.0, 3.9 and 2.7, respectively) in MOFs. The selectivity trend at a higher concentration of 50 mM was in line with the ternary breakthrough curves for 15 mM.

**Kinetic analysis of xylene uptakes in MFM-300.** To study the effect of competitive adsorption between xylene isomers and eluent (*n*-pentane) on the sorption selectivity, MFM-300(M) (M = In, V, Fe, Al) were further evaluated through liquid-phase batch adsorption experiments at 293 K using single-component solutions of three xylene isomers (0.005–1.2 M concentration range) in *n*-pentane. Comparison of the uptake curves for 15 mM solutions of *m*, *p* and *o*-xylene in *n*-pentane clearly shows that the shape of the kinetic profiles are quite different especially in the initial uptake low time region (Supplementary Fig. 15). The kinetic order is: *m*-xylene > *o*-xylene > *p*-xylene. The normalised kinetic profiles were fitted using the stretched exponential model (Supplementary Fig. 16; see "Methods"). It is evident that the adsorption of *m*-xylene and *o*-xylene follow the stretched exponential model with *β* values ~0.76 and the mass transfer constant of *m*-xylene is higher than that of *o*-xylene. The value for *β* of ~0.76 indicates the adsorption mechanism combines a surface barrier and diffusion along the pores[42,43]. However, adsorption of *p*-xylene has a *β* value of 1.6, which is outside the range of the stretched exponential model but can be fitted to the Avrami model, which confirms that adsorption of *p*-xylene is lower than *o*-xylene. The different mechanism reflects the increased length of *p*-xylene compared with *o*- and *m*-xylene and is a consequence of

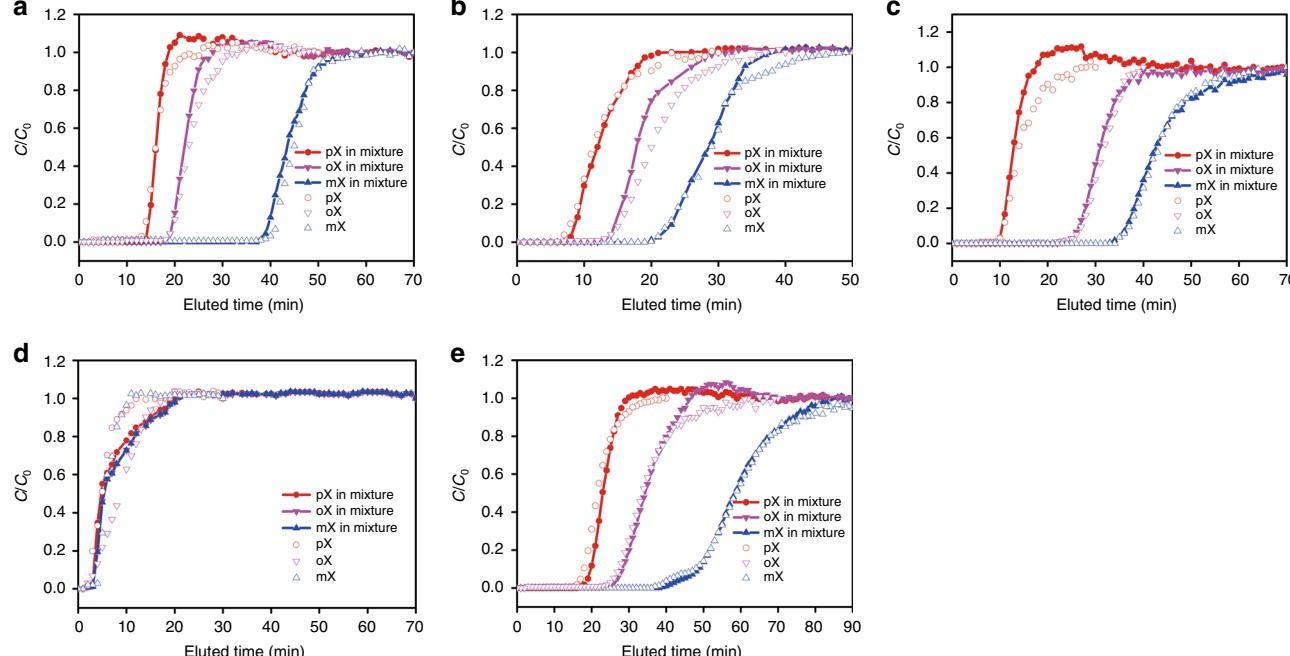

**Fig. 3 Ternary breakthrough curves for equimolar mixtures of mixture of *o*-, *m*- and *p*-xylene in different columns. a** MFM-300(In), **b** MFM-300(V), **c** MFM-300(Fe), **d** MFM-300(Al) and **e**, two columns in series comprising MFM-300(In) and MFM-300(V). Measurements were undertaken at 293 K. The solid symbol lines represent ternary breakthrough curves, which were carried out by feeding a ternary equimolar mixture of xylene isomers through the fixed-bed of MFM-300. The open symbols are the single-component breakthrough curves that were collected by feeding a single-component xylene isomer through the fixed-bed of MFM-300.

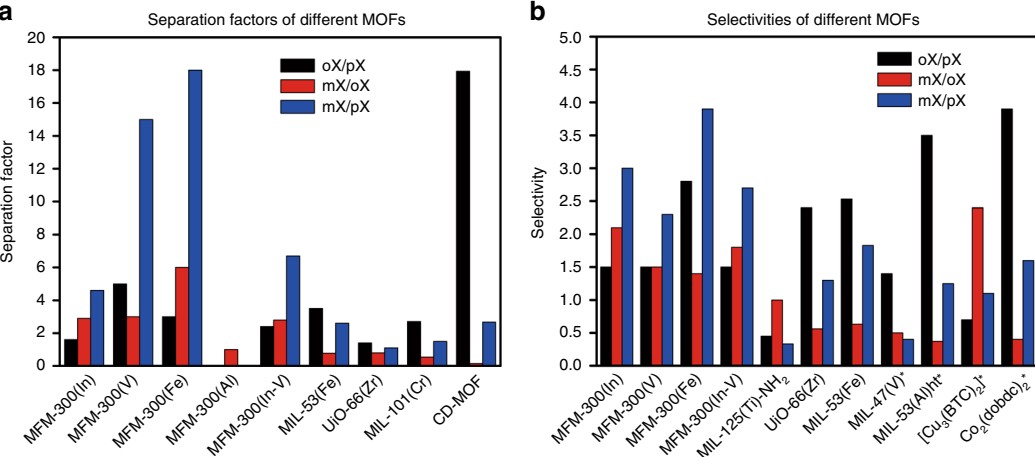

**Fig. 4 Comparison of separation performance in various MOFs. a** Comparison of separation factors calculated from chromatographic experiments; **b** comparison of selectivity data calculated from breakthrough curves. The separation factors were calculated based on pulse chromatograms using Eq. (1) in the method section and selectivities were calculated from the ternary breakthrough curves (Eqs. (3) and (4)). Selectivities denoted by an asterisk were calculated from multicomponent liquid-phase adsorption experiments. Details of conditions reported in literature examples are shown in Supplementary Table 9.

the transport of molecules through complex-shaped nanopores (Fig. 1). The adsorption of $p$-xylene on MFM-300 can also be viewed as linear over time (Supplementary Fig. 16), with a plot of $M_t/M_e$ versus time being linear ($R^2 = 0.99$) for up to 90 mins when the amount adsorbed is very close (99%) to the equilibrium uptake. It is thus evident that there are differences in the adsorption kinetics for the three xylene isomers with diffusion of species through the pore structure and different surface barriers being key factors that affect the separation of these species on MFM-300. MFM-300 maintains its xylene selectivities over a wide range of concentrations (Supplementary Fig. 17), and the selectivities calculated from the liquid adsorption data follow the same trends as in chromatographic and breakthrough experiments.

To confirm further the uptake capacity of xylene isomers in MFM-300, pure component vapour adsorption isotherms have been recorded at 318 K (Supplementary Fig. 18). The adsorption isotherms of MFM-300(M) (M = In, V) exhibit type I behaviour, indicating micropore filling by the xylene molecules. The isothermal uptakes of three xylene isomers are similar for MFM-300(In) and MFM-300(V) (2.75–2.87 mmol g$^{-1}$ and 3.45–3.88 mmol g$^{-1}$, respectively), indicating that the observed separation performance in each case primarily operates via a kinetic mechanism. In contrast, MFM-300(Al) shows a much higher uptake of $p$-xylene compared with that of $m$- and $o$- xylene (0.82, 0.45, and 0.46 mmol g$^{-1}$, respectively). The adsorption profiles correspond to a type IV isotherm, a phenomenon that has previously been observed in the adsorption of $p$-xylene in silicalite[44], where the similarity in size between the adsorbate and pore channel results in this behaviour. It is worth noting that these vapour adsorption isotherms have been measured under thermodynamic equilibrium conditions in the absence of competitive adsorption. However, the liquid-phase dynamic separation characteristics are different and involve competitive adsorption of mixtures of xylenes and solvent (i.e., $n$-pentane).

**MD modelling**. The average mean square displacement (MSD) of the centre of mass for three xylene isomers in MFM-300(In) were monitored by MD modelling (Supplementary Fig. 19, see Supplementary Movies 1–3). It can be seen that after the initial localised motion, the constant force starts to drive the molecules to diffuse along the pores. Interestingly, the transport of $p$-xylene is significantly faster than that of $m$-xylene and $o$-xylene—at 5 ps,

the MSD of $p$-xylene is much higher. These results indicate that the energy barrier for $p$-xylene molecules to diffuse inside the pores is lower than that for $m$-xylene and $o$-xylene, consistent with the separation results above and the crystal structure models of xylene-loaded MFM-300 (see below).

**Determination of binding domains for adsorbed xylene molecules within MFM-300**. The separation of xylenes via chromatographic and breakthrough experiments is governed by a combination of kinetic and thermodynamic factors. The former is regulated by pore size, whereas the latter is mediated by host–guest and guest–guest binding interactions within the pores. We sought to determine the binding domains for adsorbed xylene molecules in MFM-300(M) (M = In, V, Fe, Al) by high-resolution synchrotron X-ray powder diffraction experiments (Supplementary Tables 4–7, Supplementary Fig. 20). All xylene isomers have been successfully located in the square-shaped pores of MFM-300, and the crystal structures of xylene-loaded MFM-300 with varying host–guest interactions give rise to their unique separation properties (Fig. 5, Supplementary Figs. 21–28). The key distances of host–guest interactions are summarised in Supplementary Table 8. It is worth noting that a small amount of discrete guest water molecules are also located in the pore, residing close to the bridging -OH groups. This is likely owing to trace amounts of water in xylene or solvent or incorporation of moisture during the handling of desolvated samples.

An apparent structural distortion from $I4_122$ to $I2_12_12_1$ was observed for MFM-300(In) on loading of $p$-xylene. Interestingly, this type of distortion in MFM-300 has been predicted[45] but not observed previously. Adsorbed $p$-xylene molecules are arranged in a *zigzag* manner along the $c$ axis, and are tilted with respect to the linkers, resulting in an absence of notable host–guest π···π interactions. The distance between phenyl rings of the linker and guest $p$-xylene molecules is 5.47(5) Å, suggesting extremely weak binding interaction between $p$-xylene and the host, consistent with its rapid elution. On loading of $o$-xylene to MFM-300(In), notable π···π interactions between $o$-xylene molecules and phenyl rings on ligands are observed at a distance of 4.25(9) Å. In $m$-xylene-loaded MFM-300(In), the host–guest π···π interactions show a distance of 4.25(1) Å. In addition, there is a notable interaction between the phenyl ring of xylene molecules and the carboxyl oxygen atoms of the ligands with a distance of 4.57(1) Å,

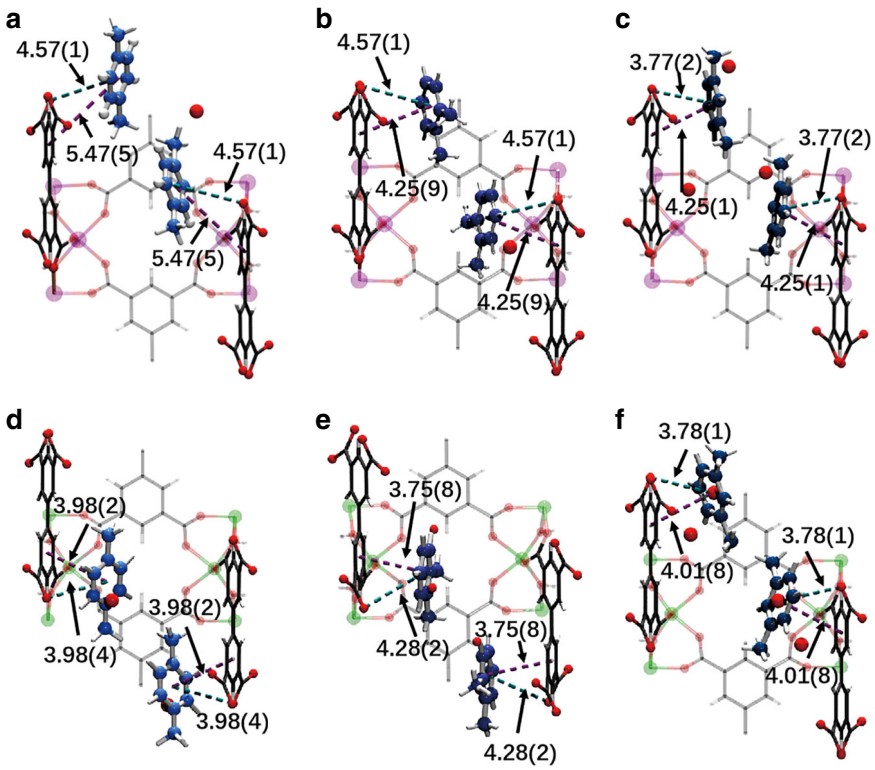

**Fig. 5 Refined xylene positions in the pores of MFM-300(In) and MFM-300(V).** All structures were solved from high-resolution PXRD data. **a** View of the structures of MFM-300(In)(*p*-xylene)$_{1.00}$, **b** MFM-300(In)(*o*-xylene)$_{0.50}$, **c** MFM-300(In)(*m*-xylene)$_{0.50}$, **d** MFM-300(V)(*p*-xylene)$_{0.50}$, **e** MFM-300 (V)(*o*-xylene)$_{0.50}$, **f**,MFM-300(In)(*m*-xylene)$_{0.50}$. The hydrogen atoms of xylene molecules have been omitted for clarity (indium, magneta; vanadium, green; carbon, grey; oxygen, red; hydrogen, white). The π···π interactions between aromatic rings of xylene molecules and ligands are highlighted in violet dashed lines; the π···O interactions between the phenyl ring of xylene molecules and oxygen atoms of carboxylic groups in linkers are highlighted in teal. For xylene-loaded MFM-300(In) the site occupancies are determined as 0.50, 0.25 and 0.25 for *p*-, *o*- and *m*-xylene, respectively. For xylene-loaded MFM-300(V) the site occupancies are determined as 0.25, 0.25 and 0.25 for *p*-, *o*- and *m*-xylene, respectively.

4.57(1) Å and 3.77(2) Å for *p*-, *o*- and *m*-xylene, respectively. Thus, the combination of host–guest interactions contributes to the strong binding of *m*-xylene, leading to its highly selective retention and hence separation in MFM-300(In).

In *p*-xylene-loaded MFM-300(V), the aromatic rings of *p*-xylene molecules and organic ligands are parallel to each other, forming π···π interactions with a distance of 3.98(2) Å. Similarly, the adsorbed *o*-xylene molecule is aligned parallel to the linker at a distance of 3.75(8) Å. In contrast, the binding of *m*-xylene to MFM-300(V) is distinct. The distance between the aromatic rings of *m*-xylene molecules and the linker is 4.01(8) Å, indicating the formation of moderate-to-weak π···π interactions. However, this is supplemented by a strong Van der Waals interaction between the aromatic ring of *m*-xylene molecule and carboxyl oxygen atom of the linker with a distance of 3.78(1) Å. Similarly, MFM-300(Fe) shows increasing strength of host–guest π···π interactions in the order of *p*-xylene, *o*-xylene and *m*-xylene with the distances between the host–guest phenyl rings decreasing along the series (see SI for details). Interestingly, MFM-300(Al) exhibits similar host–guest π···π interactions for all three xylene isomers (3.74(2), 3.53(5) and 3.53(8) Å for *p*-, *o*-, *m*-xylene, respectively) consistent with its poor separation performance in the chromatographic experiments. Thus, these guest-loaded MOF structures obtained at the host–guest adsorption equilibrium rationalise directly the observed excellent separation performance of MFM-300 materials between xylene isomers.

**Terahertz spectroscopic study of MOF-xylene binding.** Comparison of the terahertz spectra of the bare and xylene-loaded

MFM-300(In) confirms the formation of hydrogen bonds between xylene molecules and the hydroxyl bridges (Supplementary Fig. 29). Two apparent blue-shifts of the IR bands (262 cm$^{-1}$ to 268 cm$^{-1}$ and 325 cm$^{-1}$ to 332 cm$^{-1}$) have been observed in MFM-300(In) on loading of xylenes. The former is assigned as the bending mode of the carboxyl group, whereas the latter to the wagging mode of the bridging hydroxyl groups, indicating the formation of binding interactions between guest xylene molecules and carboxyl/ hydroxyl groups. Moreover, three new bands appeared at 433 cm$^{-1}$, 437 cm$^{-1}$ and 484 cm$^{-1}$, which correspond to the adsorbed *m*-, *o*-, *p*-xylene molecules, respectively[46].

In summary, refining the pore size at sub-angstrom resolution (7.4–6.3 Å in steps of 0.2 Å) in robust MFM-300(M) (M = In, V, Fe, Al) has led to precise discrimination and excellent chromatographic separation of three xylene isomers with exceptional separation factors at room temperature. The elution of the xylene isomers in MFM-300(In, V, Fe) follows the order of *p*-, *o*- and *m*-isomers, and maximises the separation of *p*- vs *m*- xylene mixtures. The tandem use of columns of MFM-300(In) and MFM-300(V) in series has achieved simultaneously optimised separation factors and chromatographic resolution, resulting in effective separation of all three xylenes. These chromatographic separations have been fully validated, and are supported by binary and ternary breakthrough experiments using continuous feeds of xylene mixtures. A combination of crystallographic and spectroscopic techniques has been applied to investigate the origins of the observed xylene selectivity in these materials. These experiments reveal that notable differences in the strength of cooperative supramolecular binding interactions with the pore

interior are directly responsible for the unusually strong binding affinity to $m$-xylene in MFM-300 materials. Thus, the underlying processes dictating the low-energy separation of xylene isomers are a complex balance of pore size, shape and chemistry, coupled with cooperative supramolecular interactions within the pore interiors and kinetic molecular sieving effects.

## Methods

**Material synthesis.** MFM-300(In), MFM-300(V), MFM-300(Al) and MFM-300 (Fe) were synthesised according to our previously reported methods[47–50]. The detailed synthesis procedures are described in Supplementary Information.

**HPLC apparatus and chromatographic separation of xylene mixtures.** A schematic view of the pulse and breakthrough reactor is shown in Supplementary Fig. 30. Approximately 1.3–1.6 g of MFM-300 sample was ground in a mortar and dispersed in a vessel with $n$-pentane in a suspension, which was then packed into a stainless steel column (15 cm length; 4.6 mm i.d.; volume 2.49 cm³) under 210 bar using a HPLC pump. The column was used for both pulse chromatographic and breakthrough experiments.

HPLC experiments were carried out in order to evaluate the ability of MFM-300 to separate for xylene isomers. All runs followed the same experimental protocol with the column packed with MFM-300 flushed with solvents for 1 h prior to the pulse experiments. Initially, the column was fed with eluent at 0.5 mL min⁻¹ at room temperature. Xylene isomers as a 1:1:1 mixture of $p$-xylene, $m$-xylene and $o$-xylene were injected into the column using a loop of 50 μL. At the same time, the fraction collector programme was started and samples taken every minute. All of the samples were analysed using a Shimadzu GC-2014 gas chromatograph with $n$-pentane, $n$-hexane and $n$-heptane used as a mobile phase.

The separation factor ($\alpha_{i,j}$) of a column for substrate $i$ and $j$ is influenced by the column packing material and the eluent used. It is a measure of the difference in interactions of two analytes with the mobile and stationary phases, and therefore the difference in retention times (Eq. (1)).

$$\alpha_{i,j} = \frac{(t_{ri} - t_m)}{(t_{rj} - t_m)} \tag{1}$$

where $t_{ri}$, $t_{rj}$ are retention times of the eluting compounds, and $t_m$ are the retention time of the given eluting compound and the bed void time, respectively. The bed void time was determined typically as the retention time of an known unretained compound, in this case, 1,3,5-triisopropylbenzene.

**Kinetic analysis of xylene uptakes in MFM-300.** A stretched exponential model has been used to analyse the differences in kinetic data for xylene uptake. This model has been used previously to describe the adsorption kinetics of a wide range of gases and vapours on metal–organic framework materials[51,52] and activated carbons[53–55]. Klafter and Shlesinger have confirmed[56] that the stretched exponential model relates the Forster direct-transfer mechanism[57], an example of relaxation via parallel channels, with the serial hierarchically constrained dynamics[58] and defect-diffusion models[59–61]. A scale-invariant distribution of relaxation times is the unifying mathematical feature of these models, and the stretched exponential model can be described by Eq. (2):

$$\frac{M_t}{M_e} = 1 - e^{-(kt)^\beta} \tag{2}$$

where $M_t$ is the amount adsorbed at certain time, $M_e$ is the amount adsorbed at equilibrium, $k$ is the mass transfer rate constant (s⁻¹) and $t$ is the time (sec). The material dependent exponent parameter $\beta$ reflects the width of the distribution of relaxation times. The stretched exponential model is three-dimensional with a single relaxation time when $\beta = 1$ (Linear Driving Force (LDF) model)[56] and one-dimensional with a distribution of relaxation times when $\beta = 0.5$. A surface barrier to diffusion of species follows the LDF model with $\beta = 1$. In the case of Fickian diffusion along the pores into a spherical particle, the value of the exponent $\beta$ is ~0.66. Values of $\beta$ between these two can be described by a model combining a surface barrier and diffusion along the pores.

**Dynamic breakthrough experiments.** In a binary breakthrough experiment, a pentane solution containing binary mixture of equimolar (40 mM) components was pumped through the column at a flow rate of 0.5 mL min⁻¹. Samples at the exhaust were collected every 1 min and analysed by GC. To explore the effect of loading, ternary breakthrough curves were conducted using equimolar ternary xylene solutions at two different concentrations (50 mM and 15 mM for each xylene monomer) following the same procedure as that used for the binary breakthrough curves. Single-component breakthrough curves were carried out using each isomer solutions at 15 mM. The adsorbed amounts $q$ (mol) were first calculated for each molecule by integration of the breakthrough curves (Eq. (3)).

$$q = \int_0^t u(C_{in} - C_{out})dt \tag{3}$$

where $u$ is the volumetric flow rate of the feed (L min⁻¹), $C_{in}$ and $C_{out}$ are the concentrations of the molecule at the column inlet and outlet, respectively (mol L⁻¹), and $t$ is the time (min). Selectivities was calculated using Eq. (4).

$$S_{i,j} = \frac{q_i}{q_j} \times \frac{C_j}{C_i} \tag{4}$$

where $q_i$ and $q_j$ are the amounts of species $i$ and $j$ adsorbed and $C_i$ and $C_j$ are the concentrations (mol L⁻¹) of species $i$ and $j$ in the external liquid phase at the column inlet.

**High-resolution synchrotron X-ray powder diffraction.** Powder samples of MFM-300 were activated by heating at 150 °C under dynamic vacuum for 1 day and sealed in vials in a glovebox. Excess amounts of pure $o$-, $m$- or $p$-xylene were injected into the sealed vials rapidly to allow the adsorption of xylene by the MOF samples over 3 days at room temperature. Xylene-loaded MFM-300 samples were dried under vacuum to remove the surface adsorbed xylenes and then loaded into 0.7 mm capillaries. High-resolution powder X-ray diffraction data were collected at room temperature on Beamline I11 at Diamond Light Source using multi-analysing-crystal detectors. Extraction of the peak positions and pattern indexation were used to complete the structural model, and final Rietveld refinements were carried out with isotropic displacement parameters for all atoms. The crystallographic sites of adsorbed xylene molecules within MFM-300 were determined by electron density peak analysis of sequential difference Fourier maps. Positions of the metal and organic linkers of the framework were fixed during intermediate Rietveld refinements, but were included in the final refinement together with the positions of the xylene molecules. Final Rietveld refinements include lattice parameters, profile and background coefficients, all atomic positions and occupancies for xylene molecules (constrained to be the same value for the atoms within each xylene molecule), and these yielded highly satisfactory agreement factors. The final refinement parameters are summarised in Supplementary Tables 2–4.

## Data availability

The data that support the findings of this study are available from the corresponding author upon reasonable request. Correspondence and requests for materials should be addressed to S.Y. and M.S. CCDC 1828516–1828527 contains the supplementary crystallographic data for this paper. These data can be obtained free of charge from the Cambridge Crystallographic Data Centre via www.ccdc.cam.ac.uk/data_request/cif.

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

## Acknowledgements

We thank EPSRC (EP/I011870), the Royal Society and the University of Manchester for funding. This project has received funding from the European Research Council (ERC) under the European Union's Horizon 2020 research and innovation programme (grant agreement no. 742401, NANOCHEM). We are especially grateful to Diamond Light Source for access to Beamlines I11 and B22. The computing resources were made available through the VirtuES and the ICE-MAN projects funded by Laboratory Directed Research and Development programme and by Compute and Data Environment for Science (CADES) at ORNL. XL, JW, NB and XZ acknowledge the financial support from Chinese Scholarship Council and the Newton Fund.

## Author contributions

X.L., J.W., N.B., C.G.M., X.H., S.X., D.W., Y.S., H.Z., K.M.T. and L.W.B.: syntheses, characterisation of MOF samples, measurements and analysis of xylene separation experiments. X.L., C.G.M., I.S., C.A.M., C.C.T., M.D.F. and G.C.: collection and analysis of synchrotron X-ray diffraction and IR spectroscopy data. Y.Q. and A.J.R.: modelling of the IR data and structures. X.L., X.Z. and T.L.: collection and analysis of the X-ray imaging data. S.Y. and M.S.: overall direction and design of project. All authors contributed to the preparation of the manuscript.

## Competing interests

The authors declare no competing interests.

**Additional information**

