## [Peer Review File · Nature Communications]

Reviewers' comments:

Reviewer #1 (Remarks to the Author):

In this report, by refining the pore size at sub-angstrom resolution in robust MFM-300(M) (M = In, V, Fe, Al), precise discrimination and excellent chromatographic separation of three xylene isomers was achieved with exceptional separation factors at room temperature. HPLC separation, breakthrough curves and batch adsorption experiments were employed to determine the separation capacity. In addition, the underlying adsorption mechanism was revealed by the Synchrotron PXRD, terahertz spectroscopic study and MD modelling. The research is novel, detailed and well-organized. The reviewer would suggest publication of this paper after all the following problems are appropriately addressed.

1. Generally, the papers related to the separation of xylene isomers will discuss the BTEX (benzene, toluene, ethylbenzene, xylenes). Compared with benzene and toluene, which can be easily separated by distillation, the ethylbenzene (E) is more difficult to separate from xylenes. Why you took B and T into consideration but excluded the E under the circumstance that E is also a by-product of catalytic reforming of crude oil?
2. Since porosity is critical for the adsorption process, all the texture properties should be concluded for these four MOFs?
3. In page 10, line 260-263, the authors stated that MFM-300 sample was ground in a mortar and packed into a stainless steel column. It is well known that the particle size would impact the separation efficiency, so what are the size of these ground particles, or how can you ensure the uniformity of the particles?
4. Although the authors stated that the structures of MFM-300 samples are rigorous, evidence such as comparisons of geometry of unit cell or pore size should be provided to clearly support this statement.

Reviewer #2 (Remarks to the Author):

In this manuscript authors have reported chromatographic separation of three xylene isomers at room temperature by using four previously reported isostructural compounds with very similar pore sizes. Experiments were performed very carefully, I don't see any technical error and manuscript is also well written. Several experiments were done and performance of the compounds are comparable/better than in few cases than the reported other compounds but at the end authors

could not find anything conclusive to design an excellent material for xylene separation, rather concluded as “it’s a very a complex balance of pore size, shape, and chemistry, coupled with cooperative supramolecular interactions within the pore interiors and kinetic molecular sieving effects.” Although details studies were done with plenty of data (which are mostly routine for such studies) but since no conclusive result was found and also compound are also previously reported, in my opinion the current manuscript is not suitable for Nat Commun.

Reviewer #3 (Remarks to the Author):

The article authored by Prof. Schröder and co-workers reports the discrimination of xylene isomers via refinement of the pore size in a series of porous metal-organic frameworks, MFM-300 (Fe, V, In, Al) at room temperature. The separation performances are confirmed by chromatography and ternary breakthrough experiments, and the separation factors and selectivity for these xylene isomers are calculated and compared with the reported MOFs and COFs. Besides, the guest loading structures are solved by the high-resolution X-ray diffraction to better understand the interaction between the host and guest. Considering the record-high selectivity of MX over PX as well as clear demonstration on discrimination mechanism, I would like to suggest acceptance after major revision as summarized below.

1. Kinetic separation can only be realized when there is significant difference in diffusivity in the pore channel between solutes. From Figure S9, I did not see any difference in diffusion rate among three xylenes. Although the authors have performed molecular dynamics simulations, which are able to obtain the self-diffusivity, determination of diffusivity of xylenes are very necessary no matter by modelling or by experiments in order to confirm so-called kinetic effect originated from pore size refinement.

2. It is not reasonable that all atoms in the MFM-300 (In) were fixed during the molecular dynamic (MD) simulation because of an apparent structural distortion of MFM-300 (In) on loading of p-xylene as mentioned in the manuscript. So, it might be no sense to compare the time of xylene molecules diffusing inside MFM-300 (In).

3. From single-component adsorption isotherms of individual xylene, the Henry coefficients are not as good as separation factor determined by chromatography. I suggest it is better to measure the vapor adsorption isotherms to further confirm their separation performance and the uptake capacity of xylene isomers.

4. What about the BET surface areas and pore size distribution of these four materials? Some basic characterizations should be included in the manuscript, such as TGA, BET surface area.

5. Figure 1, the position of xylene isomers in the channel of MFM-300 are obtained from the refined X-ray diffraction or simulated results? Figure 1e, the xylene molecules are out of the channel boundary. Please clarify.

6. Page 5, the authors mentioned that there is no change in either crystallinity or porosity of these used four materials based on the PXRD data. In fact, porosity stability could not be supported by PXRD. At least the BET experiments should be conducted to further confirm the stability and porosity of these materials.

7. Figure S5, the picture b is incomplete with a missing peak of m-xylene. In addition, as seen from the chromatograms with different flow rates (0.5 and 1.0 mL/min) on MFM-300 (In) shown in Figure a and b, the retention time of p-, m-, and o-xylene are almost the same. It is unreasonable.

8. Page 10, please add the detail information about the constant flow rate and regular intervals in the HPLC experiments section. And the separation factor was calculated using the equation 1, the original figures of HPLC should be added in Supporting Information and the t_{R} and t_{m} for MFM-300 materials should be listed, especially the t_{m} of different sample.

9. More detail information about static adsorption experiments on single-component adsorption isotherms of xylenes should be included in the experimental section. For example, Figure S9, how many samples are used for each batch for single-component liquid-phase adsorption? And what is the volume of the liquid used? Figure S10, what is the equilibrium time of each xylene isomers on these four materials for the single-component liquid-phase adsorption measurements? Additionally, the caption of y-axis of these four figures is mM/g, i.e., mmol/ml/g? It should be mmol/g. Please clarify.

10. Figure 4b, the conditions for the selectivity data calculated for various MOFs are the same as the MFM-300? If not, the detail conditions need to be added.

Reviewer #4 (Remarks to the Author):

In this communication, Yang, Schröder, and co-workers report a study on the separation of xylene isomers with the metal-organic framework family MFM-300, including the Al, In, V, and Fe versions of the MOF. The authors have completed breakthrough and dynamic separation experiments finding differences in the separation ability of the materials depending on the MOF metal cation, which is explained by the changes in the pore size arising from the differences in the metal-oxygen coordination distances.

While there are a significant number of works reported for the use of MOFs to separate xylene isomers, the performance of some of the materials here presented is highly optimal, particularly for the separation of m-xylene, making the results quite relevant.

A recent review on the separation of xylenes with porous materials that should be cited is *Ind. Eng. Chem. Res.* 2017, 56, 14725–14753 (10.1021/acs.iecr.7b03127)

- A major point of the report is focused on determining the binding sites and interaction between xylene isomers and frameworks, to account for the differences in selectivity. Thus, the authors have completed a structural analysis of the xylene loaded MOFs with the use of high-resolution powder X-ray diffraction to locate the adsorbed molecules. This is also complemented with molecular dynamic simulations.

The results of the Rietveld refinements show a nice fitting. However, the occupancy of the adsorbed xylene molecules in the provided CIFs is too high. Thus, the reported total amount of adsorbed xylene molecules ranges from 5.18 to 8 molecules per unit cell. While the molecules have partial occupancies, due to the space group symmetry, these numbers imply that xylene molecules would be bumping into each other inside the pores. Indeed, in the molecular dynamic simulations, the authors have included 8 molecules for a double volume supercell with twice c axis. The maximum loading amount should not be higher than 4 molecules per unit cell. In line with this, the resulting thermal parameters are very high, so it seems feasible from the refinement point of view that xylene molecules actually have a lower occupancy.

I am also wondering if these numbers are in agreement with the xylene sorption quantification with other methods. Have the authors quantified the amount of adsorbed xylene with other methods (such as CHN) for the samples used for the PXRD analysis?

In any case, the refinement of the occupancy factors of the adsorbed xylene molecules must be revised.

- On the other hand, according to the crystal structures, there is presence of water molecules hydrogen bonded to the bridging hydroxyl groups in the channels of some of the materials (In, Al). I guess that these molecules could not be removed even though according to the experimental details the samples have been evacuated and activated before the xylene loading and subsequent X-ray measurements. I wonder how they are actually affecting the interaction of these frameworks with the adsorbed molecules. Moreover, their presence inevitably influences the accessible pore size. In any case, their presence has been overlooked from the discussion in the binding domains section, but they are at interaction distance with the located xylene molecules, in some cases as short as 2.0 Å, so the authors should at least comment on this fact.

We thank the reviewers for their constructive comments and our responses to reviewers' comments are given in **bold italics** for clarity. The manuscript and SI have been revised according to the reviews and changes are highlighted **in yellow** for clarity.

Reviewer #1 (Remarks to the Author):

In this report, by refining the pore size at sub-angstrom resolution in robust MFM-300(M) (M = In, V, Fe, Al), precise discrimination and excellent chromatographic separation of three xylene isomers was achieved with exceptional separation factors at room temperature. HPLC separation, breakthrough curves and batch adsorption experiments were employed to determine the separation capacity. In addition, the underlying adsorption mechanism was revealed by the Synchrotron PXRD, terahertz spectroscopic study and MD modelling. The research is novel, detailed and well-organized. The reviewer would suggest publication of this paper after all the following problems are appropriately addressed.

1. Generally, the papers related to the separation of xylene isomers will discuss the BTEX (benzene, toluene, ethylbenzene, xylenes). Compared with benzene and toluene, which can be easily separated by distillation, the ethylbenzene (E) is more difficult to separate from xylenes. Why you took B and T into consideration but excluded the E under the circumstance that E is also a by-product of catalytic reforming of crude oil?

This is an excellent point. We have carried out binary breakthrough separations of E/X using MFM-300(M) (M = In, V, Fe) and the results have been added to the revised manuscript (Supplementary Fig. 13). All three materials can distinguish between E and X isomers. Combined with results shown in Supplementary Fig. 12, MFM-300(M) (M = In, Fe) show the same order of breakthrough, i.e., p-X, E, o-X and then m-X. In contrast, MFM-300(V) shows a slightly different elution sequence of p-X, o-X, E and then m-X with poor discrimination between o-xylene and ethylbenzene.

2. Since porosity is critical for the adsorption process, all the texture properties should be concluded for these four MOFs?

The adsorption isotherms and properties have been added to Supplementary Fig. 3 and Supplementary Table 1. MFM-300(M) are all microporous materials and exhibit surface areas of around 1000 m²/g, consistent with previous reports.

3. In page 10, line 260-263, the authors stated that MFM-300 sample was ground in a mortar and packed into a stainless steel column. It is well known that the particle size would impact the separation efficiency, so what are the size of these ground particles, or how can you ensure the uniformity of the particles?

This is an important point. SEM images of the ground particles have been added to Supplementary Fig. 7. The ground particles all maintained their prismatic rod shape, and the average particle size of ground MFM-300(M) (M = In, V, Fe, Al) is ca. 2, 1, 5, and 2 μm, respectively. The particle size of MFM-300(M) (M = In, V, Fe) decreases upon grinding, but MFM-300(Al) showed little change reflecting the robust nature of this material compared to its analogues.

4. Although the authors stated that the structures of MFM-300 samples are rigorous, evidence such as comparisons of geometry of unit cell or pore size should be provided to clearly support this statement.

The geometry of pore size has been added to the revised manuscript (Supplementary Fig. 1). The overall connectivity affords a porous 3D framework structure with square-shaped channels.

Reviewer #2 (Remarks to the Author):

In this manuscript authors have reported chromatographic separation of three xylene isomers at room temperature by using four previously reported isostructural compounds with very similar pore sizes. Experiments were performed very carefully, I don't see any technical error and manuscript is also well written. Several experiments were done and performance of the compounds are comparable/better than in few cases than the reported other compounds but at the end authors could not find anything conclusive to

design an excellent material for xylene separation, rather concluded as “it’s a very a complex balance of pore size, shape, and chemistry, coupled with cooperative supramolecular interactions within the pore interiors and kinetic molecular sieving effects.” Although details studies were done with plenty of data (which are mostly routine for such studies) but since no conclusive result was found and also compound are also previously reported, in my opinion the current manuscript is not suitable for Nat Commun.

We thank the reviewer for the evaluation. The complete separation of xylene isomers at room temperature is a highly challenging task. The manuscript, as pointed out by the reviewer, provides key insights for the future design of new materials requiring a complex balance between a number of parameters for any given material rather than a simple linear optimisation of any single parameter.

Reviewer #3 (Remarks to the Author):

The article authored by Prof. Schröder and co-workers reports the discrimination of xylene isomers via refinement of the pore size in a series of porous metal-organic frameworks, MFM-300 (Fe, V, In, Al) at room temperature. The separation performances are confirmed by chromatography and ternary breakthrough experiments, and the separation factors and selectivity for these xylene isomers are calculated and compared with the reported MOFs and COFs. Besides, the guest loading structures are solved by the high-resolution X-ray diffraction to better understand the interaction between the host and guest. Considering the record-high selectivity of MX over PX as well as clear demonstration on discrimination mechanism, I would like to suggest acceptance after major revision as summarized below.

1. Kinetic separation can only be realized when there is significant difference in diffusivity in the pore channel between solutes. From Figure S9, I did not see any difference in diffusion rate among three xylenes. Although the authors have performed molecular dynamics simulations, which are able to obtain the self-diffusivity, determination of diffusivity of xylenes are very necessary no matter by modelling or by experiments in order to confirm so-called kinetic effect originated from pore size refinement.

This is an excellent and critical point which is often overlooked in the literature. The reviewer is absolutely correct that there should be significant differences in the diffusivity for a kinetic separation, but there are often surface barriers to diffusion as well as to diffusion through and along pores that may give rise to kinetic effects, which lead to separation of species¹. This is the case in the current study. Thus, diffusion along the pores is not the sole aspect to be considered here. It is important to note that the diffusion mechanisms for various adsorbed substrates may also be different². Both diffusivity and the diffusion mechanisms are also a function of the amount of substrate adsorbed. Therefore, such comparisons are complex and difficult. In addition in this study, the xylene isomers are present as a solution of n-pentane, which adds the complexity of competitive adsorption with solvent.

The comparison shown in Supplementary Figure 9 (Supplementary Figure 15 in revised version) gives the uptake curves for 15mM solutions of m, p and o-xylene and this actually shows that the shape of the kinetic profiles are quite different especially in the initial uptake low time region. The kinetic order is: mX > oX > pX.

To further illustrate this point, a stretched exponential model has been used to quantify the differences in the kinetic data for xylene uptake. The stretched exponential model has been used to describe the adsorption kinetics of a wide range of gases and vapours in MOF materials^{3,4} and activated carbons⁵⁻⁷. Klafter and Shlesinger have confirmed⁸ that the stretched exponential model is a common underlying mathematical model relating the Forster direct-transfer mechanism⁹, which is an example of relaxation via parallel channels, with hierarchically constrained dynamics¹⁰ and defect-diffusion models¹¹⁻¹³. A scale-invariant distribution of relaxation times is the unifying mathematical feature of these models. The stretched exponential model is described by the following equation:

$$\frac{M_t}{M_e} = 1 - e^{-(kt)^\beta} \quad (1)$$

where M_t is the amount adsorbed at time, M_e is the amount adsorbed at equilibrium, k is the mass transfer rate constant (s^{-1}) and t is the time (sec). The material dependent exponent parameter β reflects the width of the distribution of relaxation times. This model is 3-dimensional with a single relaxation time when $\beta = 1$ [Linear Driving Force (LDF) model]⁸ and 1-dimensional with a distribution of relaxation times when $\beta = 0.5$. A surface barrier to diffusion of species follows the LDF model with $\beta = 1$. In the case of Fickian diffusion of a spherical particle along the pores, the exponent β value is ~ 0.66 . Intermediate values of β between these two can be described by a model combining a surface barrier and diffusion along the pores¹⁴. The fittings of the stretched exponential model to the normalised kinetic profiles are shown below and have been added to the revised manuscript as Supplementary Figure 16.

Supplementary Figure 16. Fitting of the stretched exponential model to the normalized kinetic profiles of a, m-xylene, b, o-xylene, c, p-xylene. d. Plot of M_t/M_e versus time based upon liquid-phase adsorption kinetics curves for p-xylene.

It is evident that the adsorption of m-xylene and o-xylene follows the stretched exponential model with β values ~ 0.76 with a higher mass transfer constant for m-xylene than for o-xylene. However, adsorption of p-xylene has a β value of 1.6 which is outside the range of the stretched exponential model and is consistent with the Avrami model. Clearly p-xylene adsorption is lower than o-xylene, but most importantly the mechanism is different. The different mechanism is probably due to the increased length of p-xylene compared with o- and m-xylene and this is a consequence of the transport of molecules in nano-sized pores, which have a complex shape. An alternative description for the adsorption of p-xylene

on MFM-300 is as linear uptake with time and this is shown in Supplementary Figure 17d. The plot of M_t/M_e versus time is clearly linear ($R^2 = 0.99$) for up to 90 min where the amount adsorbed is very close (99%) to the equilibrium uptake. Therefore, it is evident that there are differences in the adsorption kinetics for the three xylene isomers and hence the diffusion of species through the pore structure and surface barriers are the factors, which are the basis of the mechanism for the separation of these species on MFM-300. Additional discussions and references have been added to the revised manuscript.

- 1 Li, L. J. et al. Gas storage and diffusion through nanocages and windows in porous metal-organic framework $\text{Cu}_2(2,3,5,6\text{-tetramethylbenzene-1,4-disophthalate})(\text{H}_2\text{O})_2$. *Chem. Mater.* 26, 4679-4695 (2014).
- 2 Reid, C. R. & Thomas, K. M. Adsorption kinetics and size exclusion properties of probe molecules for the selective porosity in a carbon molecular sieve used for air separation. *J. Phys. Chem. B* 105, 10619-10629 (2001).
- 3 Fletcher, A. J., Cussen, E. J., Bradshaw, D., Rosseinsky, M. J. & Thomas, K. M. Adsorption of gases and vapors on nanoporous $\text{Ni}_2(4, 4'\text{-Bipyridine})_3(\text{NO}_3)_4$ metal-organic framework materials templated with methanol and ethanol: structural effects in adsorption kinetics. *J. Am. Chem. Soc.* 126, 9750-9759 (2004).
- 4 Chen, B. et al. Surface interactions and quantum kinetic molecular sieving for H_2 and D_2 adsorption on a mixed metal-organic framework material. *J. Am. Chem. Soc.* 130, 6411-6423 (2008).
- 5 Fletcher, A. J. & Thomas, K. M. Kinetic isotope quantum effects in the adsorption of H_2O and D_2O on porous carbons. *J. Phys. Chem. C* 111, 2107-2115 (2007).
- 6 Fletcher, A. J., Yüzak, Y. & Thomas, K. M. Adsorption and desorption kinetics for hydrophilic and hydrophobic vapors on activated carbon. *Carbon* 44, 989-1004 (2006).
- 7 Bell, J. G., Zhao, X., Uygur, Y. & Thomas, K. M. Adsorption of chloroaromatic models for dioxins on porous carbons: the influence of adsorbate structure and surface functional groups on surface interactions and adsorption kinetics. *J. Phys. Chem. C* 115, 2776-2789 (2011).
- 8 Klafter, J. & Shlesinger, M. F. On the relationship among three theories of relaxation in disordered systems. *P. Natl. Acad. Sci. USA* 83, 848-851 (1986).
- 9 Förster, T. Experimentelle und theoretische Untersuchung des zwischenmolekularen Übergangs von Elektronenanregungsenergie. *Z. Naturforsch. Teil A* 4, 321-327 (1949).
- 10 Palmer, R. G., Stein, D. L., Abrahams, E. & Anderson, P. W. Models of hierarchically constrained dynamics for glassy relaxation. *Phys. Rev. Lett.* 53, 958 (1984).
- 11 Glarum, S. H. Dielectric relaxation of polar liquids. *J. Chem. Phys.* 33, 1371-1375 (1960).
- 12 Bordewijk, P. Defect-diffusion models of dielectric relaxation. *Chem. Phys. Lett.* 32, 592-596 (1975).
- 13 Shlesinger, M. F. & Montroll, E. W. On the Williams—Watts function of dielectric relaxation. *P. Natl. Acad. Sci. USA* 81, 1280-1283 (1984).
- 14 Loughlin, K. F., Hassan, M. M., Fatehi, A. I. & Zahur, M. Rate and equilibrium sorption parameters for nitrogen and methane on carbon molecular sieve. *Gas Sep. Purif.* 7, 264-273 (1993).

2. It is not reasonable that all atoms in the MFM-300(In) were fixed during the molecular dynamic (MD) simulation because of an apparent structural distortion of MFM-300(In) on loading of p-xylene as mentioned in the manuscript. So, it might be no sense to compare the time of xylene molecules diffusing inside MFM-300 (In).

The MD simulation for MFM-300(In) has been re-conducted with only the metal sites fixed. The new simulation shows a similar trend as before (Supplementary Fig. 19). After the initial localised motion, the constant force starts to drive the molecules to diffuse along the pores. Interestingly, the transport of p-xylene is significantly faster than that of m-xylene and o-xylene – at 5ps, the mean square displacement of p-xylene is much higher. These results indicate that the energy barrier for p-xylene molecules to diffuse

inside the pores is lower than for m-xylene and o-xylene, consistent with experimental observations and the model discussed above. These results have been updated and included accordingly in the revised manuscript.

3. From single-component adsorption isotherms of individual xylene, the Henry coefficients are not as good as separation factor determined by chromatography. I suggest it is better to measure the vapor adsorption isotherms to further confirm their separation performance and the uptake capacity of xylene isomers.

The vapour adsorption isotherms have been measured (Supplementary Fig. 18). Supplementary Figure 18a shows the adsorption isotherms of three xylene isomers in MFM-300(In) at 318 K. All adsorbates exhibit similar uptakes of 2.75-2.87 mmol/g. The adsorption profile of each adsorbate exhibits type I behaviour, illustrating that xylene molecules are filled in the micropores of MFM-300(In). All isotherms for MFM-300(V) also exhibits type I behaviour and the uptake of three isomers is similar and in the range of 3.45-3.88 mmol/g. The adsorption of m-xylene and o-xylene in MFM-300(Al) are 0.449 mmol/g and 0.455 mmol/g, respectively, much lower than that of p-xylene (0.82 mmol/g). The adsorption profiles of m-, o-, and p-xylene correspond to a type IV isotherm, a phenomenon that has previously been observed in the adsorption of p-xylene in silicalite,¹⁵ where it was found that the similarity in size between the adsorbate and the pore resulted in this behaviour. It is worth noting that the vapour adsorption isotherms are obtained under thermodynamic equilibrium in the absence of competitive adsorption, and the separation performance can be different from dynamic separation experiments where eluent containing mixtures of xylene and solvent (in this case pentane) is in a state of flow.

15 Talu, O., Guo, C. J. & Hayhurst, D. T. Heterogeneous adsorption equilibria with comparable molecule and pore sizes. *J. Phys. Chem.* 93, 7294-7298 (1989).

4. What about the BET surface areas and pore size distribution of these four materials? Some basic characterizations should be included in the manuscript, such as TGA, BET surface area.

The BET surface areas and pore size distribution of these four materials have been added to Supplementary Table 1 and Supplementary Fig. 4, respectively. The TGA and adsorption isotherms have been added to Supplementary Figs. 5 and 3, respectively. All MOFs show decomposition at around 350 °C and a BET surface area around 1000 m²/g.

5. Figure 1, the position of xylene isomers in the channel of MFM-300 are obtained from the refined X-ray diffraction or simulated results? Figure 1e, the xylene molecules are out of the channel boundary. Please clarify.

The positions of xylene in Figure 1 are obtained from simulation using Material Studio. The Figure aims to give a direct size comparison between xylene molecules and pores of different MOF materials. In Figure 1e the xylene molecules are slightly out of the channel boundary, and this was to illustrate the “mis-match” between molecular sizes of xylene isomers and the channel of MFM-300(Al), which is consistent with vapour adsorption experiment (Supplementary Fig. 18). This has been clarified in the revised manuscript.

6. Page 5, the authors mentioned that there is no change in either crystallinity or porosity of these used four materials based on the PXRD data. In fact, porosity stability could not be supported by PXRD. At least the BET experiments should be conducted to further confirm the stability and porosity of these materials.

Adsorption experiments of used MOF materials have been conducted and compared with fresh samples (Supplementary Fig. 3 and Supplementary Table 1). The comparison shows that all used materials maintained their porosity.

7. Figure S5, the picture b is incomplete with a missing peak of m-xylene. In addition, as seen from the chromatograms with different flow rates (0.5 and 1.0 mL/min) on MFM-300 (In) shown in Figure a and b, the retention time of p-, m-, and o-xylene are almost the same. It is unreasonable.

We have double checked the raw data (Supplementary Fig. 9a and b in the revised SI) and no error was found. The retention time for different profiles are almost identical in this case, causing the overlapping of peaks in the figure.

8. Page 10, please add the detail information about the constant flow rate and regular intervals in the HPLC experiments section. And the separation factor was calculated using the equation 1, the original figures of HPLC should be added in Supporting Information and the t_{r_i} and t_m for MFM-300 materials should be listed, especially the t_m of different sample.

During the initial optimisation of separation conditions, the flow rate and regular intervals were different for different experiments. Thus, the exact values were not specified. After the optimal conditions for separation were identified, a constant flow rate of 0.5 mL/min and the sample interval of 1 min were used. The detailed information on the flow rate and regular intervals has now been added in the experimental section. The original figures of HPLC for the bed void time tests have been added in Supplementary Fig. 11 and the t_{r_i} and t_m for the MFM-300 materials have been shown in Supplementary Table 2.

9. More detail information about static adsorption experiments on single-component adsorption isotherms of xylenes should be included in the experimental section. For example, Figure S9, how many samples are used for each batch for single-component liquid-phase adsorption? And what is the volume of the liquid used? Figure S10, what is the equilibrium time of each xylene isomers on these four materials for the single-component liquid-phase adsorption measurements? Additionally, the caption of y-axis of these four figures is mM/g, i.e., mmol/ml/g? It should be mmol/g. Please clarify.

We apologise for missing these details, which have been added; please see Supplementary Figs. 15 and 17 and Supplementary section "Batch adsorption experiments" for details. Liquid phase batch adsorption experiments were carried out at room temperature in 1.8 mL glass vials filled with 0.005 g MOF and 1 mL single compound solution in pentane. For Supplementary Figure 10 (Supplementary Figure 17 in the revised SI), the equilibrium time was 2h. The y-axis has been updated to mmol/g.

10. Figure 4b, the conditions for the selectivity data calculated for various MOFs are the same as the MFM-300? If not, the detail conditions need to be added.

The conditions for the selectivity data calculated for various MOFs vary between different reports in literature. A table (Supplementary Table S9) has been added to clarify the detailed conditions.

Reviewer #4 (Remarks to the Author):

1. In this communication, Yang, Schröder, and co-workers report a study on the separation of xylene isomers with the metal-organic framework family MFM-300, including the Al, In, V, and Fe versions of the MOF. The authors have completed breakthrough and dynamic separation experiments finding differences in the separation ability of the materials depending on the MOF metal cation, which is explained by the changes in the pore size arising from the differences in the metal-oxygen coordination distances. While there are a significant number of works reported for the use of MOFs to separate xylene isomers, the performance of some of the materials here presented is highly optimal, particularly for the separation of m-xylene, making the results quite relevant. A recent review on the separation of xylenes with porous materials that should be cited is Ind. Eng. Chem. Res. 2017, 56, 14725–14753 (10.1021/acs.iecr.7b03127)

The review has been cited in ref. 5 in the revised manuscript.

2. A major point of the report is focused on determining the binding sites and interaction between xylene isomers and frameworks, to account for the differences in selectivity. Thus, the authors have completed a structural analysis of the xylene loaded MOFs with the use of high-resolution powder X-ray diffraction to locate the adsorbed molecules. This is also complemented with molecular dynamic simulations. The results of the Rietveld refinements show a nice fitting. However, the occupancy of the adsorbed xylene molecules in the provided CIFs is too high. Thus, the reported total amount of adsorbed xylene molecules ranges from

5.18 to 8 molecules per unit cell. While the molecules have partial occupancies, due to the space group symmetry, these numbers imply that xylene molecules would be bumping into each other inside the pores. Indeed, in the molecular dynamic simulations, the authors have included 8 molecules for a double volume supercell with twice c axis. The maximum loading amount should not be higher than 4 molecules per unit cell. In line with this, the resulting thermal parameters are very high, so it seems feasible from the refinement point of view that xylene molecules actually have a lower occupancy.

We sincerely apologise for the error made during the refinement of PXRD data in the original manuscript. The reviewer is absolutely correct that xylene molecules have a lower occupancy of maximum of 4 molecules per unit cell (i.e. 0.5 xylene/metal). This will achieve the complete packing of the pores of MFM-300. We have corrected the occupancy and re-conducted all Rietveld refinements accordingly. The crystallographic part has been updated and all structural figures showing the binding sites and interaction between xylene isomers and frameworks have also been updated (Figures 4 and 5, Supplementary Figs 21-28). The new refinement shows similar results (in terms of host-guest distance and binding sites) and affords the same conclusions that m-xylene interacts more strongly with the framework than the o-xylene and p-xylene, consistent with the elution sequence. Corresponding results and discussions and CIFs have been updated in the revised manuscript.

3. I am also wondering if these numbers are in agreement with the xylene sorption quantification with other methods. Have the authors quantified the amount of adsorbed xylene with other methods (such as CHN) for the samples used for the PXRD analysis?

This is a good point. TGA of bare and xylene-loaded MOFs has been conducted to quantify the adsorbed xylene (Supplementary Fig. 5). The boiling point of xylenes is around 140 °C, and the weight loss between 100 °C and 200 °C is thus due to desorption of xylene molecules. The calculated ratio between xylene molecules and metal sites in MOFs is around 0.5 for these MFM-300 materials (i.e. 4 xylene molecules per unit cell), consistent with the crystallographic study. This value cannot be determined accurately for MFM-300(V) due to the presence of a small amount of DMF, which caused a small mass loss at the same temperature range. The detailed ratios are listed below:

Ratio	MFM-300(In)	MFM-300(Fe)	MFM-300(Al)
pX per metal	0.50	0.52	0.39
oX per metal	0.64	0.55	0.46
mX per metal	0.67	0.65	0.49

4. In any case, the refinement of the occupancy factors of the adsorbed xylene molecules must be revised.

The refinement of the occupancy factors have been revised as discussed above.

5. On the other hand, according to the crystal structures, there is presence of water molecules hydrogen bonded to the bridging hydroxyl groups in the channels of some of the materials (In, Al). I guess that these molecules could not be removed even though according to the experimental details the samples have been evacuated and activated before the xylene loading and subsequent X-ray measurements. I wonder how they are actually affecting the interaction of these frameworks with the adsorbed molecules. Moreover, their presence inevitably influences the accessible pore size. In any case, their presence has been overlooked from the discussion in the binding domains section, but they are at interaction distance with the located xylene molecules, in some cases as short as 2.0 Å, so the authors should at least comment on this fact.

We apologise for missing the relevant discussion in the original manuscript, which has now been added. The reviewer is correct that these molecules cannot be removed completely even under conditions stated in the Methods section. This could be due to the presence of tiny amount of water in xylene, solvent or moisture during the handling of desolvated samples. It is also an excellent and interesting point as to how these molecules affect the host-guest interaction, which is very challenging to quantify. We believe the

host-guest interaction via the $\pi\cdots\pi$ interaction plays a key role in xylene binding. Relevant discussions have been added to the revised manuscript.

REVIEWER COMMENTS

Reviewer #1 (Remarks to the Author):

It seems that all the concerns have been appropriately addressed, so I would suggest publication of this paper in current form.

Reviewer #3 (Remarks to the Author):

The article has been well revised and some questions has been solved. But there are still some questions that needs to be answered.

1.As I mentioned before, in the revised Figure S9, the breakthrough curves with different flow rates (0.5 and 1.0 mL/min) on MFM-300 (In) in figure a and b, the retention time of p-, m-, and o-xylene are almost the same. And the caption of a-axis is the eluted volume. But how can the retention time in figure c and d (MFM-300(V)) be different with different flow rate?

2.The amount of sample is only 0.005 g for each batch for single-component liquid-phase adsorption. And the concentration of the xylene solution is very dilute. The uptake capacity was up to 3.45-3.88 mmol/g for MFM-300(V). I am wondering why the authors take such uncontrollable conditions to measure the single-component adsorption.

3.For the Table S4-7, the structural data of xylene-loaded MFM-300. The formula weight of different xylene-loaded was the same, but the formula of these materials was almost different. Please clarify.

4.Since the authors mentioned that it is very complex and difficult to compare the kinetic separation due to the surface barriers and also influenced by the amount of substrate adsorbed. But the title of this manuscript is kinetic separation of xylenes. It is better to give direct evidence to clear understand this mechanism, not just comparison of the pore size. And I did see any difference from the results of Figure S16 (normalized kinetic profiles of xylene-loaded).

Reviewer #4 (Remarks to the Author):

In this revised version, the authors have satisfactorily addressed all the raised points. I thus recommend its publication.

We thank the reviewers for their constructive comments and our responses are given in **bold italics** for clarity.

Reviewer #3 (Remarks to the Author):

The article has been well revised and some questions has been solved. But there are still some questions that needs to be answered.

1. As I mentioned before, in the revised Figure S9, the breakthrough curves with different flow rates (0.5 and 1.0 mL/min) on MFM-300 (In) in figure a and b, the retention time of p-, m-, and o-xylene are almost the same. And the caption of a-axis is the eluted volume. But how can the retention time in figure c and d (MFM-300(V)) be different with different flow rate?

Supplementary Fig. 9 summarises the results of chromatographic measurements with a one-off injection of a small amount of xylene. Chromatography shows a single peak, Gaussian or Lorentzian in shape or some other 'bell' or skewed 'bell' shaped curve. The chromatographic measurement is the time or volume corresponding to the peak maximum, and the eluted volume was used as the a-axis in this study. In contrast in the breakthrough studies, the packed bed is progressively saturated with xylene until it eventually breaks through and retention time is used for a-axis. Therefore, the conditions are different for these two techniques and some differences between the measurements are to be expected. In light to reviewer's advice, we have updated Supplementary Fig. 9 in the SI showing dual a-axes, showing both retention time and eluted volume for clarity. As the reviewer predicted, the elution time is different with different flow rate on both samples, and the lower flow rate leads to a longer elution time.

2. The amount of sample is only 0.005 g for each batch for single-component liquid-phase adsorption. And the concentration of the xylene solution is very dilute. The uptake capacity was up to 3.45-3.88 mmol/g for MFM-300(V). I am wondering why the authors take such incontrollable conditions to measure the single-component adsorption.

The dilute conditions are used to optimise the resolution of the chromatography and to avoid detector saturation. Chromatography is a very sensitive technique so optimum resolution can be achieved at lower concentrations (0-100mM in this study). Due to the low boiling point of pentane (~40 °C) the experiment was conducted in sealed GC vials (v= 1.8 mL) to minimize the loss of solvents and hence only small amounts of MOF were used.

3. For the Table S4-7, the structural data of xylene-loaded MFM-300. The formula weight of different xylene-loaded was the same, but the formula of these materials was almost different. Please clarify.

Revised. We apologise for the typos, which have been corrected.

4. Since the authors mentioned that it is very complex and difficult to compare the kinetic separation due to the surface barriers and also influenced by the amount of substrate adsorbed. But the title of this manuscript is kinetic separation of xylenes. It is better to give direct evidence to clear understand this mechanism, not just comparison of the pore size. And I did see any difference from the results of Figure S16 (normalized kinetic profiles of xylene-loaded).

This is an excellent and complex point which we have been thinking about too. As the reviewer states, there are clear differences in the initial rates in Supplementary Fig. 17. This is quantified further in Supplementary Fig. 16 by the fitting of the stretched exponential model. Supplementary Figs 17a (m-xylene) and b (o-xylene) both follow the stretched exponential model (see paper by Klafter and Schlesinger) with exponential values of 0.75 and 0.77¹, respectively. The model in the paper is limited to exponential values in the range 0.5-1.0. It is evident from Supplementary Figs 17c and 17d that p-xylene follows an Avrami model with an exponential value of 1.6 which is essentially the same equation but the exponential is > 1. Therefore, since the values of k for m-xylene and o xylene are 0.0377 and 0.0341 min⁻¹, respectively, it is fair to say that under these conditions the rates are similar, which is also consistent with the MD modelling (Supplementary Fig 19). However, the p-xylene has quite different kinetic characteristics. The data available indicate some kinetic effects. Since we can only provide limited kinetic data at this stage, the word "kinetic" has been removed from the title.

1. **Klafter, J. & Shlesinger, M. F. On the relationship among three theories of relaxation in disordered systems. P. Natl. Acad. Sci. USA 83, 848-851 (1986).**

REVIEWERS' COMMENTS:

Reviewer #3 (Remarks to the Author):

The article has been well revised and all the proposed questions has been solved. I would like to support publication.